# Towards an Accurate Faults Detection Approach in Internet of Medical Things Using Advanced Machine Learning Techniques

**DOI:** 10.3390/s22155893

**Published:** 2022-08-07

**Authors:** Mohamed Bahache, Abdou El Karim Tahari, Jorge Herrera-Tapia, Nasreddine Lagraa, Carlos Tavares Calafate, Chaker Abdelaziz Kerrache

**Affiliations:** 1Laboratoire d’Informatique et de Mathématiques, Université Amar Telidji, Laghouat 03000, Algeria; 2Facultad de Ciencias Informáticas, Universidad Laica Eloy Alfaro de Manabí, Manta 130214, Ecuador; 3Computer Engineering Department (DISCA), Universitat Politècnica de València, 46022 Valencia, Spain

**Keywords:** body sensor, cloud computing, clustering, fault detection, machine learning, WBANs

## Abstract

Remotely monitoring people’s healthcare is still among the most important research topics for researchers from both industry and academia. In addition, with the Wireless Body Networks (WBANs) emergence, it becomes possible to supervise patients through an implanted set of body sensors that can communicate through wireless interfaces. These body sensors are characterized by their tiny sizes, and limited resources (power, computing, and communication capabilities), which makes these devices prone to have faults and sensible to be damaged. Thus, it is necessary to establish an efficient system to detect any fault or anomalies when receiving sensed data. In this paper, we propose a novel, optimized, and hybrid solution between machine learning and statistical techniques, for detecting faults in WBANs that do not affect the devices’ resources and functionality. Experimental results illustrate that our approach can detect unwanted measurement faults with a high detection accuracy ratio that exceeds the 99.62%, and a low mean absolute error of 0.61%, clearly outperforming the existing state-of-art solutions.

## 1. Introduction

Patients’ supervision was and is still one of the most important preoccupations in the world and through various historical eras. In fact, with the increasing population around the world, especially in the aged population range, which is usually exposed to suffering from chronic diseases, and with the terrifying COVID-19 pandemic, this age group faces a high mortality rate.

It is known that no country has accurately determined the total confirmed number of infected people with COVID-19. All we know is that the infection status of those who have been tested. Those who have a lab-confirmed infection are, in fact, counted as confirmed cases [1]. However, according to the World Health Organization (WHO), most deaths, excluding influenza, are related to population aged over 65, representing 70.23%, and 35.509% for causes related to COVID-19 and pneumonia, respectively.

Thus, the need for remote supervision or remote health monitoring has become an urgent [2], as it allows us to:Break the transmission chain.Ensure a continuous control.Reduce health costs.

The strenuous efforts of researchers towards the fusion between the two technologies (wireless communication and sensing), has allowed us to develop WBANs (Wireless Body Area Networks), and several platforms have been launched in this direction including MEDiSN [3], CodeBlue [4], monitor HR, ECG and SpO2, Lifeguard [5], monitors ECG, breath, beat oximeter, and BP ALARM-NET [6], Medical MoteCare [7], and monitor heartbeat, SpO2, and natural parameters like the temperature.

A WBAN network consists of a few body sensors, which are characterized by their tiny sizes, and limited resources. This limitation in resources makes these electronic devices perishable and more exposed to faults, which may falsify the patient’s data, followed by a wrong health caregiver diagnosis. So, the fault detection must be taken into consideration when a WBAN is deployed; in other words, the system must be able to distinguish between the faulty measurement and the patient’s degradation state during the abnormality of the received data.

In WSNs, most existing techniques and strategies used to detect faults are based on the comparison between sensed data sent by the redundant sensor nodes. Unfortunately, this approach is not allowed in WBANs, where a few body-sensor nodes are deployed in/on/around the human body to supervise their physiologic parameters. Thus, other techniques and algorithms have been proposed to detect faults and anomalies in WBANs, such as machine learning, statistics, clustering, relay, etc. However, these proposed techniques have some limitations.

With clustering methods, we need a specific topology, as reported in [8], where extra powerful nodes are added; in other words, the infrastructure requirements are increased, and this is not desirable for two reasons: (i) it makes the patient uncomfortable, and (ii) an additional cost is introduced.

In the relay techniques, extra nodes should be added [9], making the patients uncomfortable, and the network more complex. Additionally, additional overhead processing and additional delay make it unsuitable to use for real-time data.

Statistical methods assume that our data follows a specific distribution law [10], and any deviation from this model is considered abnormal, but this is not always the case, especially when the data contain a high number of samples, requiring extra energy consumption for their processing.

As for machine learning approaches, they involve complex computation, which leads to high energy consumption and high storage capacity to store the learning data [11].

To address these aforementioned limitations, we have proposed a hybrid solution between the machine learning and the statistical techniques to detect faults in WBANs. This way we are able to achieve a novel contribution, which can be summarized as follows:The proposed solution exploits the existing correlation between the vital signs, where the focus is solely on the most correlated (strongly correlated) vital signs to the current physiological parameter, using Pearson correlation coefficients.Using the J48 decision tree, we are able to classify the received data, but in a reduced manner, using the strongly correlated vital signs that are determined in the previous phase. This way abnormal data can be properly classified.We rely on linear regression for prediction using the most correlated parameters; if the difference between the actual value and the predicted value is greater than a specific threshold, then the received abnormal value is considered faulty; otherwise, an alarm should be triggered to the health care worker for intervention. This way, storage requirements, processing energy consumption, and execution time shall be reduced.

The remainder of this paper is structured as follows: In Section 2, related works on fault detection in WBANs are presented. In Section 3, we present some essential concepts. In Section 4, we present in detail our approach. Section 5 describes and presents the experimental results. Finally, we conclude our paper in Section 6.

## 2. Related Works

Fault detection in wireless body area networks has been statistically investigated using multiple techniques such as statistical dispersion, majority voting, Markov chain model, correlation, mean, standard deviation, queue management, and a few machine learning-based solutions.

### 2.1. Statistical Solutions

Aderibigde et al. [12] used Medium Absolute Deviation (MAD) to detect temporal outliers in sensor readings, and the majority voting algorithm (VM) to identify false alarms. In [13], a link failure is detected before its occurrence using probability computation. The authors of [14] proposed a system which consists of two stages to detect fault; in the first stage they used a Pearson correlation coefficient to determine the physiological parameters strongly correlated to the actual value, and in the second stage they use statistical measures such as means and standard deviations, to obtain useful information about the capability of the actual sensor. A Markov Chain Model is also used [10] where a confusion matrix with nine states was determined. The rows represent the actual condition, and the columns describe the predicted conditions. In addition, a three-model system has been proposed: the forecasting model, the root mean square of errors, and the Markov model. In a reported work [15], a Hidden Markov Model (HMM) was used as a method for fault diagnosis of ECG sensor data. They used the Baum–Welch algorithm to estimate parameters of the HMM; then, a Viterbi-algorithm is used to check if a new sensor reading is faulty. The work in [16] illustrated that, before sending data to the PDA, a high connectivity link is chosen using probabilistic routing. In [17], the authors have used the cloud-computing hierarchy to detect sensor failures and abnormality in detected data, using historical data to calculate the average, which is then used as a decision criteria.

### 2.2. Machine Learning-Based Solutions

Machine learning techniques are also used to detect faults in WBANs. In [18], the authors used a decision tree to classify the measurements as normal and abnormal; whenever abnormal instances are located, they invoke linear regression for prediction, hence being able to discern between a faulty sensor reading and a patient entering in an emergency state. Authors in [19] used three techniques in the proposed approach: Sequential Minimal Optimization Regression (SMO regression) for the prediction of sensor values, Dynamic Threshold calculation for error computation, and Majority Voting (MV) for decision. In [20], authors used the support vector machine (SVM), which is a supervised machine learning method used for binary classification, to classify data stored in a data set source. Whenever an abnormal class is detected, the linear regression is invoked to predict the value being compared with the actual one. In the work [11], a hybrid solution was proposed between a machine learning SVM classifier and a nature-inspired optimization named lion hunting algorithm for fault detection in WBANs. In [21], the authors considered three body sensors—HR, BP, and SpO2—and they used a Bayesian network to estimate the conditional probability; whenever the value of this probability is greater than a specific threshold, then the sensor reading is diagnosed as correct. Otherwise, the sensor reading is diagnosed as faulty.

In [22], the Naive Bayesian Network classifier is used in a co-existence environment (multiple WBANs). In [23], an artificial neural network is used along with linear regression to predict values to be compared with the actual ones, from one side, and the associative neural networks but in an enhancement way, by considering the cascade feed-forward propagation, from another side. An embedded self-healing for detecting and recovering faults in WBANs has been proposed [24], where the autonomic-computing paradigm is adopted using spine2 and the KNN classifier. In [9], the relay-based technique is used where extra nodes are established to overcome the degradation or the problem of battery depletion. However, sometimes the relay technique is coupled with the network coding technique.

The authors of [25] used relay techniques with two other techniques: network coding and hierarchical modulation. The network coding it is used for property working direct link, and if a coded link is not available, then, the communication switches to the hierarchical modulation, so that the channel is not completely deteriorated. The Middleware-based approach has been proposed in [26], where authors integrated a software tool between the transport and MAC layers to handle some channel impairments.

Figure 1 summarizes the distribution rates of each explored technique for fault detection in WBANs.

To avoid the drawbacks of the above mentioned solutions, we propose in this paper a new solution combining both statistical and machine learning techniques to ensure a higher accuracy. Our proposal encompasses three phases: the first phase is the correlation phase, where we use Pearson correlation to focus only on the most correlated vital signs according the actual one. Afterwards, the classification phase is used to classify the sensed data, and for all data classified as abnormal, we use linear regression to retrieve the correct value. If the difference is less than a predefined threshold, an alert is triggered as this value may present an urgent situation of the monitored patient’s health, otherwise the actual value is faulty.

## 3. Background

In this section, we will present some basic concepts based upon which our proposed is defined. We start by providing an overview of the decision tree, which is a machine learning tool, used for the classification. Afterward, we present the linear regression used for the prediction.

### 3.1. Decision Tree

This is a machine learning tool used to build a binary classification model where the decision is binary (true/false, normal/abnormal, …). It is considered as a graphical representation for this classification model, where nodes represent tests on attributes, edges represent an outcome of the tests, and leaves represent classes.

In our case, the nodes are the physiological parameters, and the leaves are the two classes (normal or abnormal). To build this decision tree, there are two main notions used—the entropy and the information gain—that are modeled using Equations (1) and (2), respectively.
(1)$$Entropy\left(X\right)=\sum_{i=1}^{c}\alpha_{i}\log_{2}\alpha_{i}$$
where *c* is the number of classes, and $\alpha_{i}$ is the probability of each class.
(2)$$G\left(X,P_{k}\right)=E\left(X\right)-\sum \frac{|P_{ki}|}{\left|X\right|}E\left(P_{ki}\right)$$
where $P_{ki}$ is the value of the parameter $P_{k}$ at the time *i*; for more details please refer to [27].

In the literature we can find several algorithms to construct decision Trees, including: ID3, C4.5, CART, CHAID and MARS.

#### J48 Classifier

J48 is an open source Java implementation of the C4.5 algorithm. C4.5 creates a decision tree based on a set of labeled input data based on the Iterative Dichotomiser 3. J48 works according to Algorithm 1.   
**Algorithm 1:** J48 Algorithm.Stage1: The leaf is labeled with a similar class if the instances belong to similar classes.Stage2: For each attribute, the potential data will be figured, and the gain in the data will be taken from the test on the attribute.Stage3: Finally, the best attribute will be chosen depending on the current selection parameter.

### 3.2. Linear Regression

Regression analysis is a statistical technique used to predict a dependent variable from one or more independent variables, and can be mathematically described through the following equation:(3)$$y_{im}=\beta_{0}+\beta_{1}x_{i1}+\ldots +\beta_{k}x_{ik}$$
where $y_{im}$ is the dependent variable, and $x_{ik}$ are the independent variables, also called regressors, and:(4)$$\beta_{j}=cov\left(X_{j},Y_{j}\right)/var\left(X_{j}\right)$$

## 4. Our Contributions

In this section, we provide details concerning our contribution. In this regard, we start by presenting the system model. Then, we proceed by discussing the proposed method, and the different phases it encompasses: correlation, classification and prediction.

### 4.1. System Model

The adopted scenario is shown in Figure 2, where our WBAN contains *N* body sensors $\left(BS_{1},BS_{2},\ldots ,BS_{N}\right)$. Theses BSs are deployed in, on, or around the patient’s body to supervise its physiological parameters. Then, the sensed data are sent in wireless mode to the PDA (a smartphone for instance), which is characterized by a large memory, long battery life, and high computational capabilities. This PDA processes the gathered data, and then sends the data to the health caregiver through a public network via an access point. We can denote by a vector the measured parameters at a specific time *t* as: $X_{t}=\left(VS_{1t},VS_{2t},\ldots ,VS_{Nt}\right)$, where $VS_{it}$ is the measurement of the *i*th vital sign at time *t*, by the body sensor $BS_{i}$. Combining all the data, we can obtain the matrix:$$X=\left|\begin{array}{c} X_{1}\\ X_{2}\\ .\\ .\\ .\\ X_{M} \end{array}\right|$$

### 4.2. Proposed Method

In the real world of medicine, there is a correlation between the physiological signals of the human body [28]. Thus, any change that appears on one parameter is somehow associated to the level change of other correlated parameters.

In our contribution, we have exploited the correlation between these vital parameters, and our objective is to reduce the resource consumption of our WBAN. We have used the decision tree classification and the linear regression prediction in an optimized way, unlike the existing approaches, by integrating the correlation phase so that it solely focuses on the most correlated vital signs, as illustrated in Figure 3.

Our fault detection system works as follows: when the PDA receives data, the data are passed through the first component in our FDS, which is the correlation phase, to determine the most correlated physiological parameters; this phase determines the set of most correlated vital signs. The decision tree is then used (j48 in our experimentation) to build a classification model, hence allowing us to properly classify the current data. If the classification is judged as abnormal, we then invoke the third component, which is the linear regression component, to predict the corresponding value. After that, a comparison is achieved. If the difference between the actual value and the predicted value is greater than the specific threshold (10%), the actual value is then considered as faulty; otherwise, it triggers an alarm to the health caregiver for intervention.

#### 4.2.1. Phase 1: Correlation

Our system, when receiving data related to the current vital signal, determines the most correlated parameters based on the Pearson correlation; as an example, we focus on PULSE and RESP (Resperation rate) as the most correlated vital signs of HR (Heart rate).

Specifically, this correlation is determined using the following formula:(5)$$r\left(vs_{i},vs_{j}\right)=\frac{n\left(\sum vs_{i}vs_{j}\right)-\left(\sum vs_{i}\right)\left(\sum vs_{j}\right)}{\sqrt{\left[n\sum vs_{i}^{2}-{\left(\sum vs_{i}\right)}^{2}\right]\left[n\sum vs_{j}^{2}-{\left(\sum vs_{j}\right)}^{2}\right]}}$$
where $r(vs_{i},vs_{j})$ is the Pearson’s correlation coefficient between the two vital signs, $vs_{i}$ and $vs_{j}$.

This way, we can get the following correlation matrix:$$Corr\_Matrix=\left(\begin{array}{c} vs_{11},vs_{12}\\ vs_{21},vs_{22}\\ \vdots ,\vdots \\ vs_{n1},vs_{n2} \end{array}\right)$$

This correlation phase operates according to the following diagram illustrated in Figure 4.

Notice that it uses the correlation matrix constructed previously so as to solely focus on strongly correlated vital signs. These correlated vital signs should be used to build the decision tree classifier for classification, and the linear regression for prediction. This way the classification model can be reduced, and the linear regression prediction model becomes simpler, as shown in the following equation:(6)$$HR_{i}=\beta_{0}+\beta_{1}PULSE_{i}+\beta_{2}RESP_{i}$$

#### 4.2.2. Phase 2: J48 Decision Tree Classifier

After determining the most correlated vital signs according to the actual parameter, our system can build a classification model using the second component(J48) decision tree classifier, where the intermediate nodes represent the values of the vital signs, these nodes are mapped to the leaves which represent classes (normal/abnormal) through edges.

#### 4.2.3. Phase 3: Linear Regression

When the received signal or the actual physiological parameter are classified as abnormal, we invoke our third component, linear regression, which is a statistical model to predict the corresponding value based on a reduced set of parameters; then, a comparison should be achieved between the actual and the predicted values, as described in our fault detection Algorithm 2 see below.   
**Algorithm 2:** Proposed fault detection algorithm.
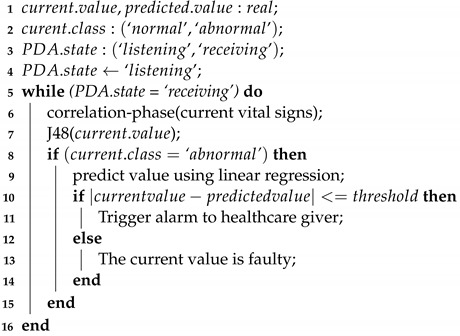


Our algorithm summarizes the three phases, in the first stage and according to the actual vital sign, we determine the most correlated vital signs, after that the actual value is injected in the classification model to determine its class, if the class is abnormal, then the linear regression is used for the comparison. In other words, the regression model is used to distinguish between the sensing data faults, and the data representing a real health-related critical situation.

## 5. Experiment and Results

To validate our approach, we conducted our experiments on the same data set, and with the same machine learning tool, used by several existing approaches in this literature, where a dataset is extracted from the real clinic database Physionet BANK [29] through the PhysioBANK ATM window (see Figure 5). More precisely, we used the 221n record that contains 7 Vital signs and 3516 instances. Afterwards, faults are randomly injected before triggering the classification process through WEKA data mining tool. We also assume that the different body sensors are interconnected following a star topology.

This record contains 7 vital signs, as shown in Table 1. We used WEKA to compare our approach with the works in [19], KNN, and Naïve Bayesian Network.

The variation of the ABPmean, HR, PULSE, RESP, and SpO2 signals are shown in Figure 6.

Besides, the vital signs used to construct the decision tree are: ABPmean, HR, and PULSE; the resulting decision tree, as generated by WEKA, is shown in Figure 7.

For comparison, we used different evaluation metrics. Concerning the kappa coefficient, it allows comparing the classification models with the standard classifier ZERO-Rules. We also employed the Mean Absolute Error (MAE), which measures the error between the predicted and the actual values according to the following formula:(7)$$MAE=\frac{\sum_{i=1}^{n}|pv_{i}-av_{i}|}{n}$$
where $pv_{i}$ is the *i*th predicted value, $av_{i}$ is the *i*th actual value, and *n* is the number of the instances.

Accuracy detection is also used as an evaluation criteria, and it is calculated according to the following equation:(8)$$ACC=\frac{TP+TN}{TP+TN+FP+FN}$$
where TP refers to true positives, TN refers to true negatives, FP refers to false positives, and FN refers to false negatives.

Finally we use the ROC curve that plots the true positive rate against the false positive rate and the obtained results.

Concerning the kappa coefficient, we observe that our approach can achieve the best rate, as illustrated in Figure 8.

Concerning classification errors, the mean absolute error criteria is presented in Figure 9, where we can see that the smallest errors occur in our approach. In Figure 10, the accuracy detection is presented, and we found that our approach is the most accurate among the four models.

At this stage, it is worth mentioning that the actual accuracy value is achieved through the variation of the selected threshold; hence, each time we change the threshold value, we achieve a certain accuracy. To avoid these redundant and recurrent changes and operations, the ROC curve is used, as it represents the true positive rate against the false positive rate, as described by the two following equations:(9)$$TPR=\frac{TP}{TP+FN}$$
(10)$$FPR=\frac{FP}{FP+TN}$$

We know that these numbers are described by the confusion matrix that is listed at the bottom of each model, where the rows are the actual values, and the columns are the predicted values. Figure 11, shows the ROC curves for our approach and the other three approaches, where the largest area is in our approach plotted under 1, the second is KNN plotted under 0.9867, the third is Naïve Bayesian network plotted under 0.9127 and the last is Haque et al. [19] plotted under 0.8321. Thus, our approach is highly accurate and outperforms all other alternatives, as it achieves the highest TPR, and the lower FPR values, as presented in Table 2.

Concerning the linear regression running time, our approach takes 0.14 s to execute, outperforming the work by Haque et al. [19] where the time overhead for that same task was of 0.83 s.

## 6. Conclusions and Future Work

Currently, WBANs are receiving increasing attention due to their great potential at achieving faster and more accurate diagnostics of patients, especially when a remote diagnosis is required. However, WBANs are characterized, on the one hand, by the lack of resources and the importance of the medical information of the patient that is sent; on the other hand, such information can be subject to errors or changes in its values, which may lead to a wrong diagnosis. This may cause deterioration or degradation of the patient state, and so detecting faults is of utmost importance.

In this research work, we address the challenge of fault detection in WBANs. Since this problem is considered as a classification problem, we have proposed a hybrid approach between machine learning tools and statistical techniques, where the J48 classifier is used to build a classification model, and when the classification result is judged as abnormal, we invoke the linear regression for prediction; in addition, we achieve this in an efficient manner by taking advantage of the existing correlation between the vital signs (using Pearson correlation). The experimentation evidenced that our approach is more accurate than several existing models and approaches, being able to achieve a detection accuracy rate of 99.62%, and a minimal absolute error of 0.61%, outperforming other state-of-the-art solutions.

As a future work, we plan to use blockchain technology as a mean to achieve a better traceability of the whole vital signs detection process, along with edge computing to reduce the overall latency of the system.

## Figures and Tables

**Figure 1 sensors-22-05893-f001:**
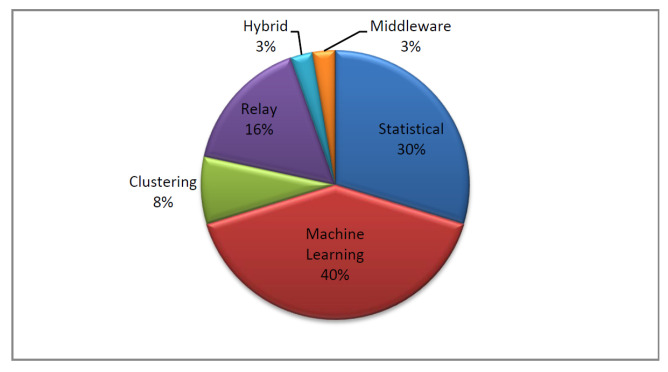
Distribution rates of the used techniques.

**Figure 2 sensors-22-05893-f002:**
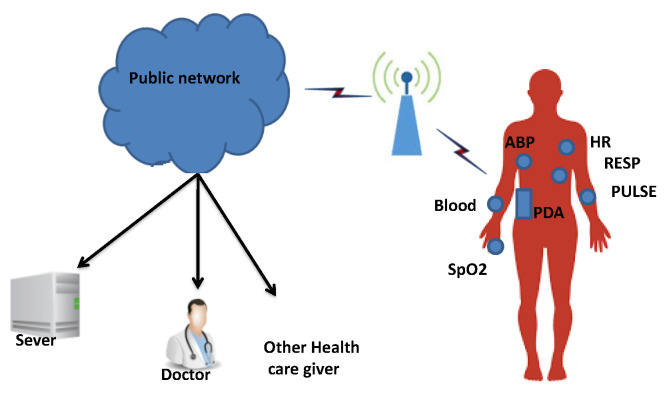
WBAN architecture overview.

**Figure 3 sensors-22-05893-f003:**
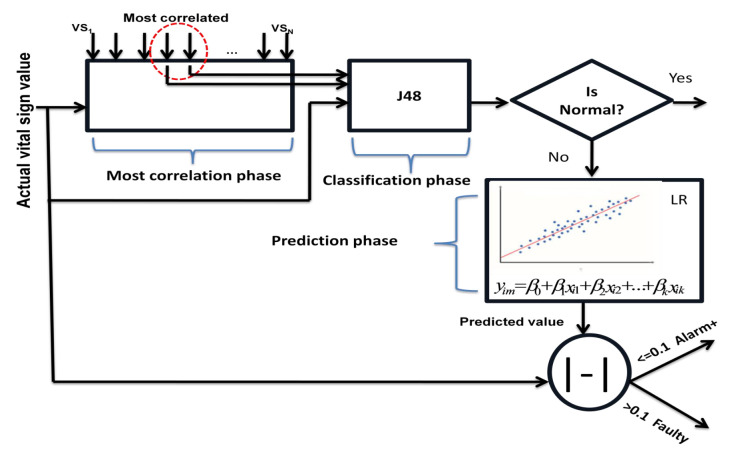
Our fault detection system in WBANs.

**Figure 4 sensors-22-05893-f004:**
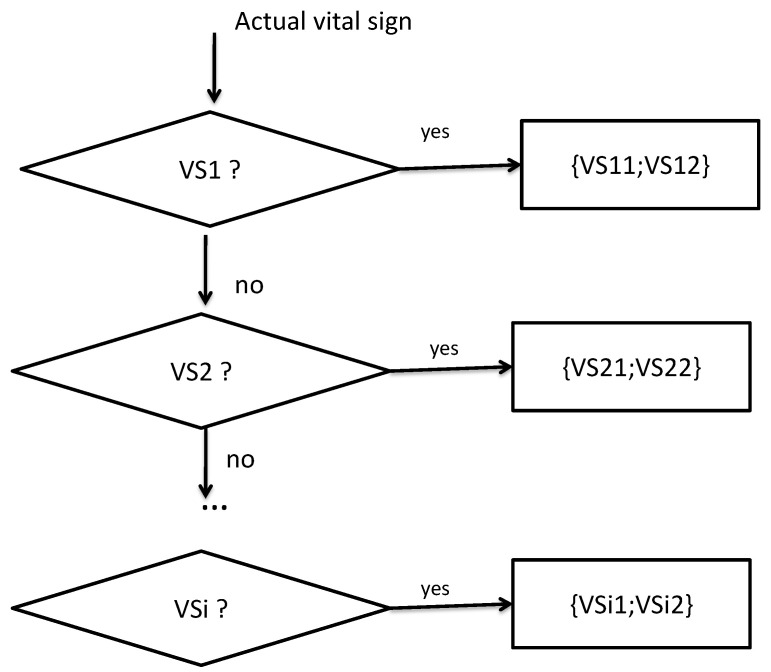
Correlation phase workflow.

**Figure 5 sensors-22-05893-f005:**
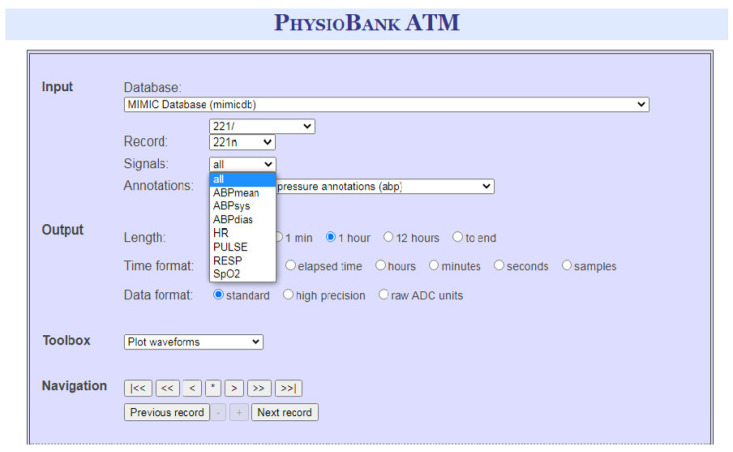
PhysioBANK ATM window.

**Figure 6 sensors-22-05893-f006:**
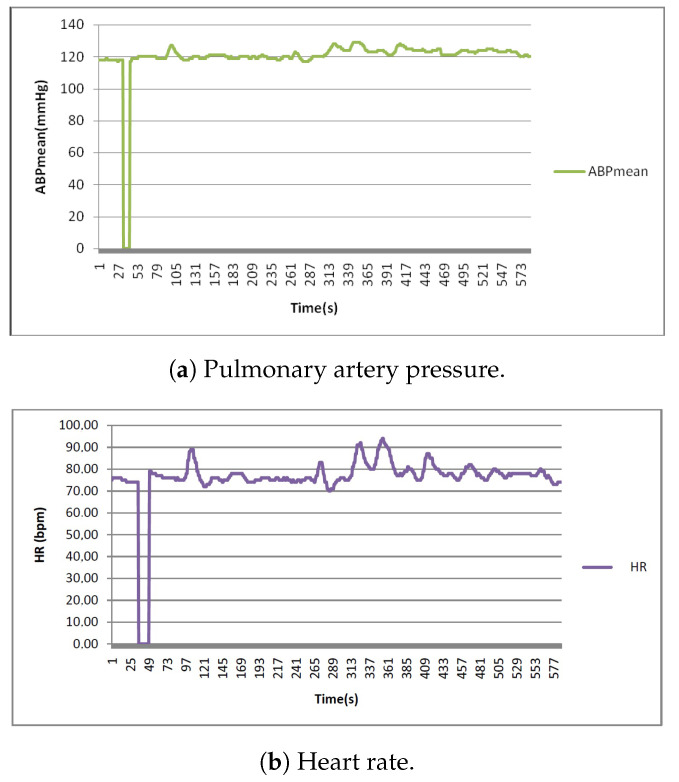
Vital signs.

**Figure 7 sensors-22-05893-f007:**
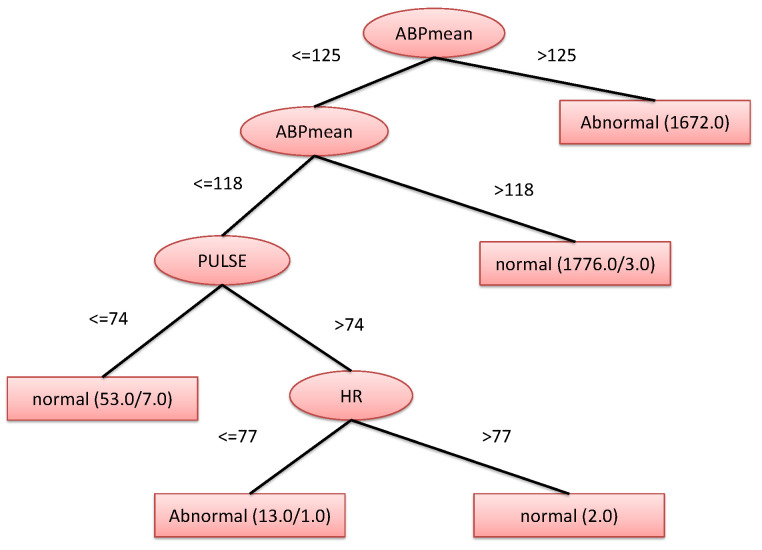
Resulting decision tree.

**Figure 8 sensors-22-05893-f008:**
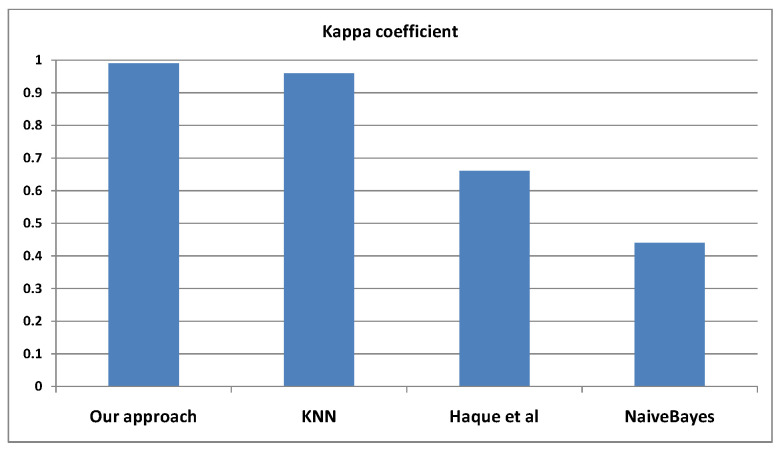
Kappa statistics for our approach when compared to 3 relevant alternative approaches in the literature.

**Figure 9 sensors-22-05893-f009:**
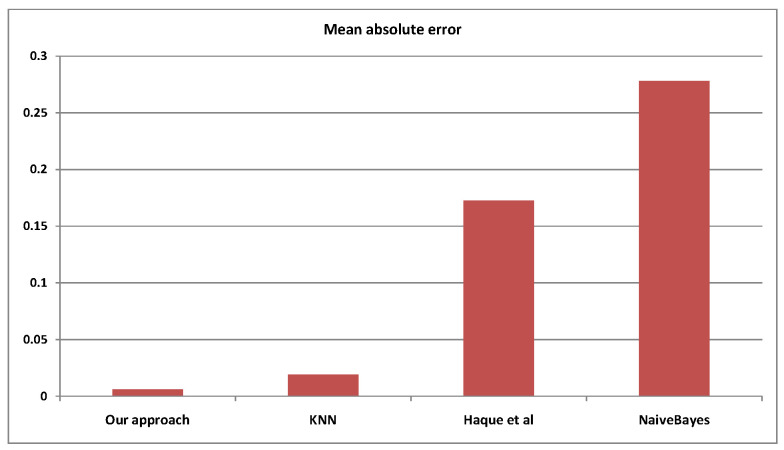
Mean absolute error for our approach when compared to 3 relevant alternative approaches in the literature.

**Figure 10 sensors-22-05893-f010:**
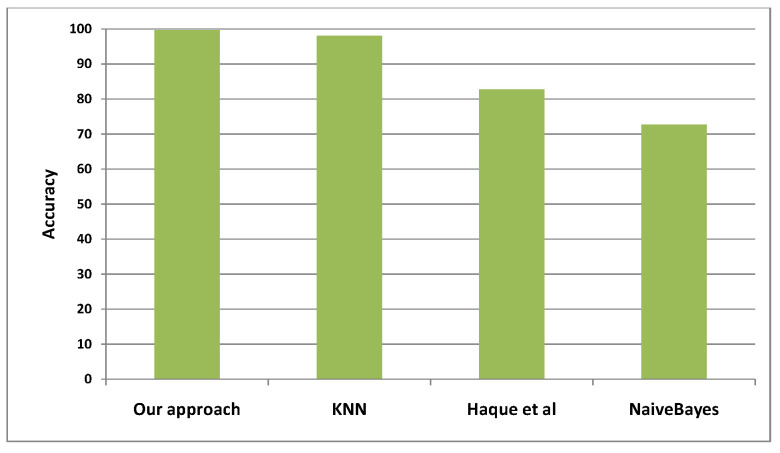
Detection accuracy for our approach when compared to 3 relevant alternative approaches in the literature.

**Figure 11 sensors-22-05893-f011:**
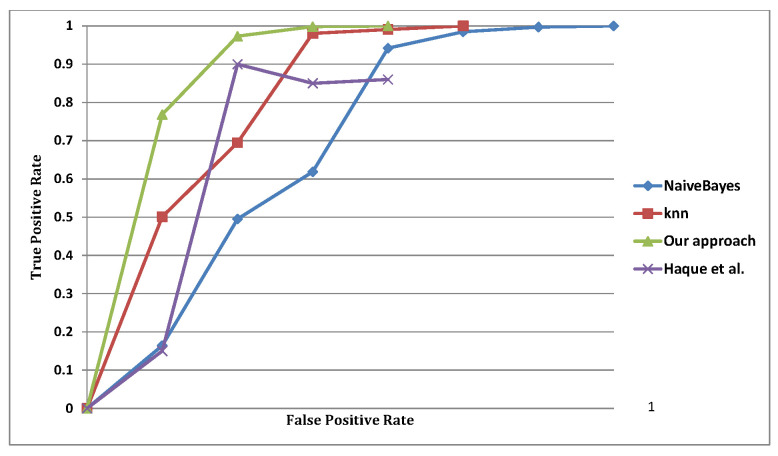
Proposed solution’s ROC curve.

**Table 1 sensors-22-05893-t001:** Experiment parameters.

Parameter	Value
Clinic Bank	PhysioBank ATM
DataBase	MIMIC
Record	221n
Signal1	ABPmean
Signal2	ABPsys
Signal3	ABPdias
Signal4	HR
Signal5	PULSE
Signal6	RESP
Signal7	SpO2

**Table 2 sensors-22-05893-t002:** TPR and FPR for each approach.

Rate	Our Approach	KNN	Haque et al. [19]	NaiveBayes
**TPR**	99.80%	98.10%	98.10%	97.10%
**FPR**	0.60%	2.2%	21.6%	53.6%

## Data Availability

Not applicable.

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
