# Peer review of "Towards an Accurate Faults Detection Approach in Internet of Medical Things Using Advanced Machine Learning Techniques"

_sensors, 2022, doi:10.3390/s22155893_

Round 1

Reviewer 1 Report

The authors propose the hybrid method to detect faults in WBAN devices. The hybrid method is combined one of the machine learning and statistical methods. The following concerns should be resolved before publication.

1) (Decision Tree) The authors propose utilization of the decision tree to determine abnormality. The decision tree is formed using the most correlated vital signs. The authors should present the vital signs to form the decision tree and most importantly, the resulting decision tree.

2) (Consistent Variable) In Section 4, the measured vital sign is expressed repeatedly using different variables: P in Line 205, VS in Figure 3, and v in (6). Those variables should be edited to be consistent in the manuscript.

3) (Database) The authors describe that the database from [31] was used for experiment in the manuscript. The authors should provide information how many data sets were used to obtain the decision tree and the results in Figures 7 to 9.

4) (Hybrid model) The authors propose the combination of the decision tree and the regression model to distinguish the faulty and alarming statuses. However, the decision tree can be used to directly distinguish the faulty and alarming statuses. What benefits exist in using the regression model?

5) (Lengthy and short parts) Sections from 3.1 and 3.4 need to be removed in the manuscript because those describe general information that can be found in many references. Instead, Section 3.5 should be described more details about the Decision Tree algorithm, i.e., J48.

Author Response

First of all, we would like to thank the reviewers for the valuable comments on the original version of this paper. We have incorporated all suggested changes in this new version, which we think has now adequately addressed all the comments, with appropriate explanations.

In this document, we detail the changes introduced, summarizing, in boldface, the issues pointed out by the reviewers. All the changes have been incorporated into the revised version of our original manuscript. 

1.       (Decision Tree) The authors propose the utilization of the decision tree to determine abnormality. The decision tree is formed using the most correlated vital signs. The authors should present the vital signs to form the decision tree and most importantly, the resulting decision tree 

-Following the reviewer comment, we now presented the correlated vital signs that are: ABPmean, HR, and PULSE. Figure 7 is also added to show the resulting decision tree. 

2.      (Consistent Variable) In Section 4, the measured vital sign is expressed repeatedly using different variables: P in Line 205, VS in Figure 3, and v in (6). Those variables should be edited to be consistent in the manuscript.

-We sincerely thank the reviewer for this detailed review. We adequately revised our manuscript to harmonize all occurrences of the different variables.

3.      (Database) The authors describe that the database from [31] was used for the experiment in the manuscript. The authors should provide information how many data sets were used to obtain the decision tree and the results in Figures 7 to 9.

-The used datasets are extracted from Physionet BANK ATM database. More precisely, we used the 221n record that contains 7 Vital signs and 3516 instances. Afterwards, faults were randomly injected before triggering the classification process through the WEKA data mining tool. We included these missing details in the new version of the paper.

4.      (Hybrid model) The authors propose the combination of the decision tree and the regression model to distinguish the faulty and alarming statuses. However, the decision tree can be used to directly distinguish the faulty and alarming statuses. What benefits exist in using the regression model?

-We sincerely thank the reviewer for giving us the chance to clarify this point. In our approach, we start by the classification phase to classify the sensed data, as normal or abnormal. If the sensed data is classified as abnormal, we use linear regression to predict the corresponding value. Afterwards, we compare the actual value with the predicted value and, if the difference is higher than a predefined threshold, the received abnormal value is considered as faulty. Otherwise, an alarm should be triggered to the healthcare support as this may represent a critical health situation. In other words, the regression model is used to distinguish between the sensing data faults and the data representing a real health-related critical situation.

5.      (Lengthy and short parts) Sections from 3.1 and 3.4 need to be removed in the manuscript because those describe general information that can be found in many references. Instead, Section 3.5 should be described more details about the Decision Tree algorithm, i.e., J48.

-Following the reviewer recommendation, we removed the sections from 3.1 to 3.4 in the background. We also extended the description of the decision tree, presented the different types, and presented the J48 decision tree algorithm that we used in our work.

Finally, we sincerely thank the reviewers for their detailed and constructive reviews; all suggested modifications have been taken into account in this new version of the paper.

Reviewer 2 Report

-What machine learning method was used?

-Figure 6 is not visible.

-Figure 8: What represents the Oy axis?

-What type of temperature, HR, PULSE, and SpO2 sensors were used?

-How are interconnected in WBAN these sensors? 

Author Response

First of all, we would like to thank the reviewers for the valuable comments on the original version of this paper. We have incorporated all suggested changes in this new version, which we think has now adequately addressed all the comments, with appropriate explanations.

In this document, we detail the changes introduced, summarizing, in boldface, the issues pointed out by the reviewers. All the changes have been incorporated to the revised version of our original manuscript. 

1.       What machine learning method was used?

-In our work, we used two machine learning methods: Decision Tree and Linear Regression, which are a supervised learning techniques. We have updated several parts of the paper to clearly state this point.

2.      Figure 6 is not visible

-Following the reviewer comment, we now improved the quality and the resolution of Figure 6. 

3.      Figure 8: What represents the Oy axis?

-The Oy axis represents the mean absolute error. We updated the figure 8 by including this important missing information. Thank you for this detailed and very constructive review. 

4.       What type of temperature, HR, PULSE, and SpO2 sensors were used?

-The body sensors for this study can be Telosb motes, or Shimmer.

5.      How are interconnected in WBAN these sensors? 

-The body sensors are interconnected following a star topology.

Finally, we sincerely thank the reviewers for their detailed and constructive reviews; all suggested modifications have been taken into account in this new version of the paper.

Round 2

Reviewer 1 Report

All my concerns have been resolved.